# Exploring Relationships among Organizational Culture, Empowerment, and Organizational Citizenship Behavior in the South Korean Professional Sport Industry

**Yunduk Jeong** [1] , **Euisoo Kim** [2] , **Minhong Kim** [3] **and James J. Zhang** [4,*]

1   Department of Sport Management, Kyonggi University, Suwon 16277, Korea; fcgangwon@nate.com
2   Department of Kinesiology, University of Georgia, Athens, GA 30602, USA; euisoo.kim25@uga.edu
3   Department of Kinesiology, Health Promotion and Recreation, University of North Texas, Denton, TX 76203, USA; minhong.kim@unt.edu
4   Department of Kinesiology, University of Georgia Athens, GA 30602, USA
*   Correspondence: jamesz48@uga.edu

**Abstract:** The purpose of this study was to examine the structural relationships among organizational culture, empowerment, and organizational citizenship behavior (OCB) of professional sports organizations in South Korea. In particular, this study emphasized the mediating effect of empowerment on the relationship between organizational culture and OCB. Research participants were 606 employees affiliated with 42 professional sports teams. The validity and reliability of the involved measures were examined through conducting confirmatory factor, Cronbach's alpha, and correlation analyses. A structural equation modeling analysis with maximum likelihood estimation was conducted to test the relationships among the research variables. The findings revealed that all of the sub-factors of organizational culture (i.e., clan culture, adhocracy culture, and market culture), with the exception of hierarchy culture, were positively influential of perceived empowerment, which was in turn positively influential of OCB. The path coefficients were statistically significant. The findings further revealed that perceived empowerment partially or fully mediated relationships between the sub-factors of organizational culture and OCB. Unlike previous studies, our study focused on studying organizational culture at a specific managerial level, an underdeveloped area of research in sport management. In particular, the findings of this study contribute to sport management practices by uncovering the mediating function of empowerment on the relationship between organizational culture and OCB, indicating the importance of empowering employees when managing professional sports organizations.

**Keywords:** organizational culture; empowerment; organizational citizenship behavior; professional sport; sustainability management

## 1. Introduction

Although one of the main purposes of owning and operating a professional sport franchise is to make profits through various marketing activities at home and abroad, a majority of professional sport clubs in South Korea rely heavily on the financial support from their parent companies and municipal governments, which make up between 60% and 80% of the teams' finances [1]. The inception of South Korean professional sports leagues in the early 1980s is uniquely attributable to the government's desire to divert people's political attention [2]. Three professional sports leagues were introduced simultaneously between 1982 and 1983 (i.e., professional baseball in 1982, soccer in 1983, and ssireum—traditional Korean wrestling—in 1983), and large corporations such as Samsung and Hyundai were

demanded to comply with the government's request to invest a significant amount of financial resources to create and run these teams [3]. Essentially, professional sports teams have been primarily tasked with promoting the corporate image, marketing products, or carrying out corporate social responsibility initiatives instead of generating profits as an independent entity [4]. Given the unique organizational environment within which Korean professional sports teams operate, do-nothingism and immorality are widespread among employees. Some HR staff have been skeptical that the promotions and compensations for front office employees are based on one's ability; instead, they are often based on cronyism [1].

Recently, however, many large corporations and municipalities have reduced their financial support for professional sports teams. For example, the Samsung Professional Baseball Team had an annual budget from its parent company, which had been reduced from approximately USD 37 M in 2017 to USD 30 M in 2018 [5]. There are two main reasons for this descending trend. First, parent companies perceive that utilizing professional sports teams as promotional tools are less effective than they were portrayed in the past [1]. Second, parent corporations put more emphasis on marketing communications via various social media, which allow direct, active communication with potential consumers [1]. Despite this change, labor costs for administrative employees continue to make up a large portion of sport clubs' operational budget. For instance, Football Club Seoul spent approximately USD 5 M on labor, which was approximately 20% of the subsidy from its parent corporation, resulting in a net loss of $537,991 in 2018 [5]. As most professional sports teams in South Korea are running a similar deficit, these teams cannot afford to hire more employees. To cope with the challenge, the top management of a team often promotes a reduction of human resources by improving work efficiency. This has placed a burden on current employees to work extra hours without additional compensation, which is in fact a widely accepted norm in Korean society over the last few decades of fast economic growth [2].

Moreover, globalization and advanced technology have forced the industry to reckon with increased competition against renowned overseas sports leagues to secure domestic sport fans [6]. To comply with these changes and resolve managerial issues, Korean professional sports teams expect their employees to be proactive in dealing with limited financial and human resources [1]. In other words, employees working for Korean professional sports teams are often required to handle a variety of added responsibilities and concentrate on more than just their job-related tasks that are aimed to boost productivity and efficiency. Cooperative and collaborative activities among employees that require them to perform beyond their formal job descriptions and work duties are indispensable [7]. In this regard, individuals' voluntary efforts and willingness to cooperate with other employees are deemed essential for efficient and effective operation. Katz and Kahn [8] indicated that if an organization consists of employees who are devoted only to their personal roles, the organization will likely perish. In other words, employees must be able to perform proactive extra-role behaviors to help the organization when it is necessary to attain desired results [9].

The extra-role behaviors that are required for superior performance are also known as organizational citizenship behavior (OCB) [10]. Previous research findings highlighted the positive contribution of OCB on superior performance of organizations, as OCB encourages employees to spontaneously participate in and dedicate themselves to organizational issues, both of which enhance the efficiency and performance of an organization without additional spending [11]. Given its complexity and service-oriented nature, the success of sports organizations largely depends on their ability to adapt appropriately to the rapidly changing environment [12]. Specifically, a consumer-oriented attitude, flexibility, and wide task coverage of employees, as well as a cooperative working environment, were identified as critical factors that contribute to the success of sports organizations [1]. To enable these behaviors, sports organizations should share their vision with employees and guarantee a certain degree of autonomy to their employees to effectively and efficiently achieve organizational objectives and goals.

Organizational culture has attracted considerable attention as a means for companies' long-term survival and productivity improvement in a rapidly changing modern society [13]. Although much

organizational culture research has been conducted to date, most studies have centered on the unique management styles and behaviors in the U.S., Japanese, and Western European contexts [14]. This has produced certain study limitations, including the ambiguity of the concept of organizational culture, the lack of measurement tools, and insufficient empirical inquiry [15]. As organizational culture has been identified as an effective tool to carry out OCB [16], assessing the influence of organizational culture on OCB is necessary to guarantee the achievement of organizational objectives and goals in Korean professional sports teams. In addition, in the process of maximizing OCB, the autonomous participation and commitment of employees in their assigned job tasks and other organizational activities are required [16,17]. To achieve these goals, empowerment has gained attention since empowerment refers to "a change (internal or external to the person) is an increase in empowerment if (if and only if) it is an increase in the person's control over the determinants of his/her quality of life, through (necessarily) an increase in either health (e.g., through self-confidence, self-esteem, self-efficacy, autonomy), or knowledge (self-knowledge, consciousness raising, skills development, competence), or freedom (negative or positive)" [18]. Thus, empowerment is a way to maximize organizational change and performance by eliminating the powerlessness that has been widespread in the U.S. and by making employees more engaged in their work [1].

Although numerous attempts have been made to investigate and verify the relationship among organizational culture, empowerment, and OCB, limited research has been conducted in a unique context where distinctive and different organizational structure, culture, customs, and values exist [19]. As Korean professional sports teams possess these unique operational characteristics, investigating the underlying dynamics of OCB and its determinants could enhance existing literature and produce practical implications and insights to those who share similar characteristics or face organizational issues. Considering the unique features of Korean professional sports teams, the purpose of this study was to assess the influence of organizational culture on OCB and the mediating effect of empowerment on the relationship between organizational culture and OCB (see Figure 1).

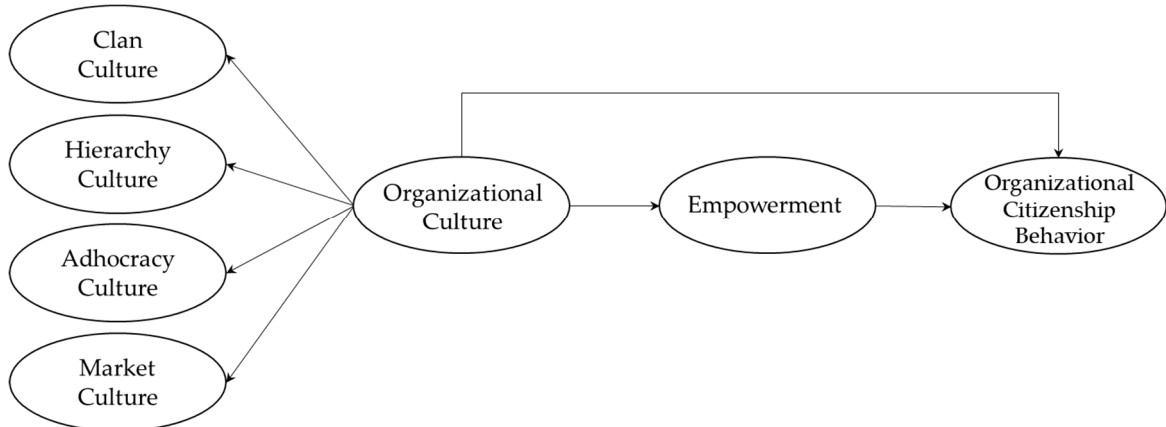

**Figure 1.** Conceptual framework.

## 2. Theoretical Background, Research Hypotheses, and Model

### 2.1. Organizational Citizenship Behavior

In order for an organization to function effectively, at times, it is critical that employees must not only perform well on their given tasks but also take initiatives to engage in extra-role behaviors. Previous research has confirmed that having employees who take on tasks beyond their normal job duties is imperative for organizational effectiveness and efficiency [20]. This notion—known as organizational citizenship behavior (OCB) [21]—is generally defined as an "individual behavior that is discretionary, not directly or explicitly recognized by the formal reward system, and that in the aggregate promotes the effective functioning of the organization" [22]. Previous literature has

emphasized that OCB is critical to managing an organization as the organization may be vulnerable (i.e., not be responsive enough) to social changes when employees only focus on their assigned explicit role-behaviors [8]. More specifically to the sports setting, OCB has been identified as a cohesive and driving force for the benefit of sports organizations [23,24].

Numerous scholars have attempted to better understand the multidimensional nature of OCB. For instance, Van Dyne et al. [16] proposed a framework with three underlying dimensions (i.e., social participation, loyalty, and functional participation), while Organ [22] developed a five-factor OCB model containing altruism, conscientiousness, sportsmanship, courtesy, and civic virtue, which has become the most widely adopted framework in the OCB literature. Following suit, this study applied Organ's [22] framework as Organ's [22] altruism and courtesy were imbedded into social participation while sportsmanship and civic virtue overlap with loyalty within Van Dyne et al.'s [16] framework. Particularly, the current research focused heavily on two of Organ's [19] sub-concepts—altruism and civic virtue—since putting more emphasis on these allowed us to better capture the unique characteristics of Korean professional sports teams [25].

## 2.2. Organizational Culture

Culture is a broad and complex concept that each discipline defines differently. In social science, it is commonly defined as "the principal attitudes, behaviors, values, beliefs, and customs that typify the functioning of a society" [26]. Therefore, culture works as a common frame of reference that encourages cultural group members to share similar thoughts, attitudes, emotions, and behavior, and to leads them to perceive and evaluate their surroundings in a similar manner [27]. The influence of organizational culture on its members works similarly as it is referred to as a "complex set of values, beliefs, assumptions and symbols that define the way in which a firm conducts its business" [15]. In other words, organizational culture can be perceived as an organizational DNA, which typifies the behaviors of members and organizations that influence heavily on organizational capacity.

When it comes to cultural studies, one of the most extensively cited research guidelines is Hofstede's framework [27], which proposes that national culture can be categorized based on four bipolar dimensions: individualism-collectivism, masculinity-femininity, power distance, and uncertainty avoidance. Based on this categorization, the culture of Korean society values group membership (collectivism), affiliation (femininity), acceptance of hierarchical order (large power distance), and avoidance of behavioral differences (strong uncertainty avoidance). These cultural traits of Korean society also heavily influence the organizational culture of many Korean corporations including professional sports teams, since organizational culture would be formulated based on national culture, particularly in Korea. For instance, a large number of Korean public organizations embrace hierarchy culture [28], and the culture of most private corporations in Korea was characterized by hierarchy and market culture [29]. In addition, since professional sports organizations in South Korea are not established with a vision to make a profit for sustaining or thriving as aforementioned, the managerial culture of a majority of professional sports teams in South Korea is rather primitive, dependent, and lack of proactive mechanisms for long-term sustainability.

Considering the importance of organizational culture for explaining organizational behavior, previous studies have attempted to distinguish specific types of organizational culture that lead to higher organizational effectiveness [1]. Among the various typologies of organizational culture, the Competing Value Framework (CVF) is one of the most widely adopted theoretical foundations for not only analyzing the characteristics of organizational culture [30] but also analyzing the effects of organizational culture on organizational effectiveness [13,30]. This model identified four types of organizational culture: clan culture, hierarchy culture, adhocracy culture, and market culture.

Clan culture emphasizes cooperation among members, including factors such as a family-like atmosphere, morale, communication, and cohesion, while focusing heavily on maintaining and improving human relationships within the organization [31]. Hierarchy culture describes control, hierarchy, compliance with rules and regulations, and role sharing within formalized and structured

organizations [13]. Adhocracy culture involves creativity, innovation, and challenge as ways to organizational success [13], which also lead to the acquisition of new resources and entrepreneurship. Finally, market culture refers to an attempt to increase the competitiveness of the organization against the external environment by centering on productivity (i.e., goal achievement, performance-driven, and positive competition among colleagues) as a core focus [32].

As organizational culture significantly influences employees' behaviors, it is highly likely that there is a positive relationship between organizational culture and OCB. Indeed, Kar and Tewari [33] identified that heightened OCB was observed among employees who were well aligned with and understood the organizational values and ethics. They also suggested that support, structure, and risk tolerance were some of the most critical factors of organizational culture that significantly influence the OCB of employees. In addition, OCB can be influenced by open commitment, mutual trust, and degree of shared values among employees of an organization [16]. Considering such relationships, some scholars went a step further to assess the influence of various types of organizational culture on OCB. For example, Kerr and Slocum [34] examined the influence of two different types of organization culture (i.e., clan and market culture) and found that employees who shared a clan culture exercised better OCB than those with a market culture. However, market culture also positively influences the OCB of employees. Choi, Cho, and Hong [35] examined the relationships among organizational culture, job characteristics, and OCB and empirically demonstrated that market culture has a positive impact on the conscientiousness and civic virtue of OCB, while hierarchy culture influenced altruism, which is a pertinent dimension of OCB. Moreover, Ryu and Ryu [36] found supporting evidence that adhocracy culture is a predictor of OCB in Korean and Chinese enterprises. Therefore, the following hypotheses were developed for the current study:

**Hypothesis H1.** *Organizational culture will positively influence OCB.*

**Hypothesis H1-1.** *Clan culture will positively influence OCB.*

**Hypothesis H1-2.** *Hierarchy culture will positively influence OCB.*

**Hypothesis H1-3.** *Adhocracy culture will positively influence OCB.*

**Hypothesis H1-4.** *Market culture will positively influence OCB.*

Organizational culture is one of the critical components that facilitates employee empowerment. More empowered employees have reported a stronger belief in their self-efficacy, ability, and influence in the decision-making process for the success of the organization [37–40]. Foster-Fishman and Keys [41] described the process of empowerment as an interaction between an individual and the environment and proposed that organizational culture plays an important role in empowering employees and achieving organizational success. For instance, clan culture was identified as an antecedent of empowerment. Jeong [1] suggested that when a sports organization's culture values cooperation, communication, and teamwork (i.e., clan culture), individuals tend to be empowered. Im, Yoon, Son, Nam, and Jang [42] investigated the impact of organizational culture on empowerment in the context of nursing and proposed a positive relationship between hierarchy culture and empowerment. The authors elaborated that since working environments in hospitals require strong discipline and obedience of rules and regulations, employees felt empowered and proud of themselves when they strictly followed a given role and order.

Previous research found that when an organizational culture values flexibility (i.e., adhocracy culture), the organization is more likely to give decision-making authority to employees, thereby increasing productivity and performance [43]. Further, employees tend to be more empowered when the culture of an organization properly assesses the values of their individual contributions and independence [44]. More importantly, Spreitzer [45] found that a participative climate was one of five

factors (i.e., role ambiguity, span of control, sociopolitical support, access to information and resources, and participative unit climate) that positively influenced employees' feelings of empowerment. In other words, if an organizational culture encourages employees' active participation in their task-related decision-making process, employees are more likely to feel that they are important assets to the organization, thus facilitating empowerment. Accordingly, it is hypothesized that:

**Hypothesis H2.** *Organizational culture will positively influence empowerment.*

**Hypothesis H2-1.** *Clan culture will positively influence empowerment.*

**Hypothesis H2-2.** *Hierarchy culture will positively influence empowerment.*

**Hypothesis H2-3.** *Adhocracy culture will positively influence empowerment.*

**Hypothesis H2-4.** *Market culture will positively influence empowerment.*

*2.3. Empowerment*

Initially, the term empowerment was regarded as a way of increasing political and economic power to improve the living conditions of ethnic minority and socially marginalized people in the U.S. [46]. Although there is no consensus on the definition of empowerment, empowerment became an important topic in business as a tool to maximize organizational change and performance by eliminating the "powerlessness" and by making employees more engaged in their workplace. Thomas and Velthouse [47] defined empowerment as an "increased intrinsic task motivation" (p. 666) and suggested that the concept is multifaceted and thus should employ several factors to uncover the core concept of empowerment: sense of impact, competence, meaningfulness, and choice. Building on Thomas et al. [47], Spreitzer [48] conceptualized the four components of psychological empowerment that measure the active orientation of employees' work roles—meaning, competence, self-determination, and impact.

Meaning is the value of an occupation goal or purpose that is determined based on an employee's own ideals or standards [47] or how employees harmonize their values, beliefs, and behaviors with the given task role within an organization [49]. As empowerment is a mindset of employees on their role in the organization [50], acknowledging the meaningfulness and significance of an employee's role and task to achieve goals and strategies of organization provides employees with incentives to perform well in the future [51]. Competence involves an individual's belief in his or her ability to perform a skillful activity [52], and previous research identified a positive relationship between competence and organizational performance [53–55]. Particularly, inferior job performance was observed among individuals with low job competence, who tried to avoid trying new tasks and stuck solely to their routine jobs [53]. Self-determination refers to experiencing "a sense of choice in initiating and regulating one's own actions" [56]. In the work environment, self-determination is closely linked to work autonomy as it gives employees the authority to make decisions. In this regard, workers who are vested with more authority for work-related decisions, or have more work-related self-determination, were found to be better performers [57,58], as they possessed a strong sense of ownership over their tasks and also personal responsibility for and commitment to organizational outcomes [47]. Finally, impact is the extent to which individuals believe they can influence strategic, administrative, or operating outcomes at work [37]. When employees believe that the performance of their job is irrelevant to the outcome of an organization, they are less likely to perform better [37]; however, effectiveness of job performance was found to be higher when employees had a sense of the "impact" of their job on the overall organizational performance [55].

As empowerment mitigates employees' feelings of powerlessness and leads them to engage more actively in their work, previous research provides supporting evidence of a positive impact

of empowerment on employees' OCB [59–61]. By exploring the relationships among empowerment, perceived organizational support, OCB, job embeddedness, and job performance among fast food service workers, Karavardar [60] identified the importance of empowerment by revealing the positive influence of empowerment on OCB. Lee et al. [61] also confirmed the positive relationship between empowerment and OCB in a sport setting by revealing the significant impact of empowerment on OCB that also mediated the relationship between transformational leadership and OCB. Considering such relationships, the following hypotheses were developed:

**Hypothesis H3.** *Empowerment will positively influence OCB.*

To the best of our knowledge, no previous studies have tested the mediating effect of empowerment on the relation between organizational culture and OCB. However, some studies suggest the probable mediating role of empowerment between the two constructs. For example, empowerment is often considered a key intervening construct between organizational culture and dedication to an organization. Yu, Jung, and Choi [62] investigated a sports organization in South Korea and showed that organizational culture directly impacts the dedication of employees to the organization while culture indirectly influences OCB through empowerment. Moreover, Gwak [63] found a full mediating effect of empowerment on the relationship between the hierarchical culture and effectiveness of organizations. Likewise, Kim and Kim [64] showed that the effectiveness of an organization is directly affected by both organizational culture and empowerment; meanwhile, empowerment was also found to play an intervening role between culture and effectiveness. Based on these findings, the following hypotheses were formulated.

**Hypothesis H4.** *Empowerment will mediate the relationship between organizational culture and OCB.*

**Hypothesis H4-1.** *Empowerment will mediate the relationship between clan culture and OCB.*

**Hypothesis H4-2.** *Empowerment will mediate the relationship between hierarchy culture and OCB.*

**Hypothesis H4-3.** *Empowerment will mediate the relationship between adhocracy culture and OCB.*

**Hypothesis H4-4.** *Empowerment will mediate the relationship between market culture and OCB.*

## 3. Method

### 3.1. Participants and Procedure

To investigate the impact of organizational culture on both OCB and empowerment, as well as the mediating effect of empowerment on the relationship between organizational culture and OCB, the data were collected by using a purposive sampling technique from front office employees at 42 professional sports teams in South Korea, including 10 baseball teams, 20 soccer teams, seven basketball teams, and five volleyball teams. These surveys were administered by both on-site and postal-mail procedures. The researchers contacted the managers of the front offices, and the questionnaires were distributed upon agreement. An informed consent form was included at the beginning of the questionnaire for compliance with the Institutional Review Board's (IRB) protocol. An on-site survey was the preferred protocol of data collection. When visits were limited, survey questionnaires were mailed to the front office managers. A total of 651 respondents completed a paper-and-pencil self-administered questionnaire; however, only 606 questionnaires were returned with completion, including 342 from the on-site process and 293 from the postal-mail process. Demographic characteristics included basic personal information, such as gender (71.5% male and 28.5% female); age (31.8% were in their 20 s, 48.5% were in their 30 s, and 19.7% were in their 40 s or older); administrative department (36.1% public relations and marketing, 32.1% management support, 20.6% team (athletes) operation and support, and 11.2% other departments); and level of education (5.8% high school, 8.2% associated degree, 74.9% university, and 11.2% graduate).

*3.2. Instrumentation and Data Analyses*

To measure organizational culture, the CVF was adopted and revised from Cameron et al. [13] and Lee, Shiue, and Chen [65], which contained a total of 16 items, with four items for clan culture, hierarchical culture, adhocracy culture, and market culture, respectively. In the case of empowerment, four items were adopted and modified from Spreitzer [48] and Hochwälder and Brucefors [66]. Finally, four items from Organ [21], Podsakoff, MacKenzie, Moorman, and Fetter [67] and Park et al. [25] were utilized to measure OCB. Particularly, items assessing altruism and civic virtue were included in the questionnaire by considering the unique characteristics of South Korean professional sports organizations. All items were measured by using a 5-point Likert scale, ranging from 1 (strongly disagree) to 5 (strongly agree). After the initial questionnaire was formulated, the questionnaire was sent to a panel of experts (i.e., seven panels included sport management and statistics professors and professional sports team managers) for the test of content validity in terms of item relevance, representativeness, and clarity. With minor edits, all of the items were retained based on the standard of 80% agreement among the panel members [68].

Descriptive statistics (e.g., means and standard deviations, skewness, kurtosis, sociodemographic characteristics, etc.) were calculated by using the latest version of the Statistical Package for the Social Sciences (SPSS). A confirmatory factor analysis (CFA) was conducted to assess the goodness of fit of the measurement model by using the latest version of AMOS software. Normed chi-square ($\chi^2/df$), root mean square error of approximation (RMSEA), standardized root mean square residual (SRMR), and comparative fit index (CFI) were adopted to assess the goodness of fit. Reliability estimates (i.e., Cronbach's alpha scores and composite reliability [CR]) and validity estimates (i.e., factor correlation and average variance extracted [AVE] values) were calculated. Next, the proposed theoretical framework was tested by employing a structural equation modelling (SEM) analysis.

## 4. Results

*4.1. Preliminary Analyses*

First, a CFA was employed to assess the dimensionality of the six-factor measurement model (i.e., clan culture, hierarchical culture, adhocracy culture, market culture, empowerment, and OCB) by using the maximum likelihood estimation. The fit indices for the initial six-factor measurement model with 24 items were not satisfactory ($\chi^2/df$ = 5.039, RMSEA = 0.082, CFI = 0.896, and SRMR = 0.056). Considering the initial results of the CFA, a model re-specification was made due to poor factor loadings and high modification indices [69]. Consequently, a total of five items were removed (i.e., two items under hierarchical culture and one each under market culture, empowerment, and OCB). A follow-up CFA with a total of 19 items showed an acceptable fit. The normed chi-square ($\chi^2/df$ = 4.311) exceeded the suggested cut-off value of 3.0 [70]; however, other fit indices were in the acceptable ranges (RMSEA = 0.074, SRMR = 0.039, and CFI = 0.94).

In terms of instrument reliability, the Cronbach's alpha for the six factors scored higher than the suggested cut-off value of 0.70 [71], ranging from 0.80 (hierarchical culture) to 0.89 (clan culture and OCB), and CR estimates also exceeded the recommended cut-off value of 0.70 [72]. For the validity of the instruments, all factor loadings were satisfactory and correlations among the six factors were less than 0.85. The AVE values were higher than the threshold of 0.50 [73] establishing discriminant validity of the measure (see Table 1).

**Table 1.** Summarized Result for Reliability and Validity Assessments.

| Items | λ | α | CR | AVE |
|---|---|---|---|---|
| **Clan Culture** | | | | |
| Our organization has a family atmosphere. | 0.80 | | | |
| Our organization has high morale. | 0.75 | 0.89 | 0.89 | 0.68 |
| Our organization regards cooperation among colleagues as important. | 0.88 | | | |
| Our organization value communication between the upper and lower classes. | 0.86 | | | |
| **Hierarchy Culture** | | | | |
| Our organization values the hierarchy. | 0.93 | 0.80 | 0.82 | 0.69 |
| Our organization has a pecking order between the upper and lower classes. | 0.73 | | | |
| **Adhocracy Culture** | | | | |
| Our organization emphasizes the challenges of new things. | 0.76 | | | |
| Our organization collects various opinions of employees in dealing with the task. | 0.73 | 0.85 | 0.85 | 0.59 |
| Our organization values the recruitment of new employees. | 0.76 | | | |
| Our organization emphasizes new ideas and creativity. | 0.82 | | | |
| **Market Culture** | | | | |
| Our organization values the goal. | 0.83 | | | |
| Our organization evaluates employees based on their performance and quality. | 0.75 | 0.83 | 0.83 | 0.62 |
| Our organization stresses maximum performance in the given circumstances. | 0.78 | | | |
| **Empowerment** | | | | |
| The work I do is meaningful to me. | 0.82 | | | |
| I am confident about my ability to do my job. | 0.90 | 0.89 | 0.90 | 0.75 |
| I have significant autonomy in determining how I do my job. | 0.78 | | | |
| **Organizational Citizenship Behavior** | | | | |
| I help my absent colleagues work well. | 0.91 | | | |
| I am willing to help my colleagues with high workloads. | 0.92 | 0.89 | 0.90 | 0.75 |
| I voluntarily participate in meetings that I consider important. | 0.76 | | | |

*4.2. Hypothesis Testing*

To analyze the relationships between the constructs and test hypotheses, an SEM analysis and multiple path analyses were employed. The structural model with a second-order organizational culture indicated a satisfactory fit to the data ($\chi^2/df$ = 4.88, CFI = 0.92, RMSEA = 0.08, and SRMR = 0.06). Second-order organizational culture had a significant influence on empowerment ($\gamma$ = 0.93) and OCB ($\gamma$ = 0.96), and empowerment was also shown to have a significant impact on OCB ($\gamma$ = 0.78). When the direct impact of each organizational culture sub-factor was tested using path analyses, we found that all sub-factors (clan culture, adhocracy culture, and market culture), except for hierarchy culture, positively influenced empowerment ($\gamma$ = 0.43, 0.41, and 0.13, respectively; see Figure 2). Therefore, H2-1, H2-3, and H2-4 were supported. Path coefficients from organizational culture sub-factors to OCB also revealed positive influences (clan = 0.43, hierarchy = 0.15, adhocracy = 0.30, and market = 0.11), supporting H1-1, H1-2, H1-3, and H1-4. Finally, the influence of empowerment on OCB was statistically significant as well ($\gamma$ = 0.78), supporting H3 (see Figure 2).

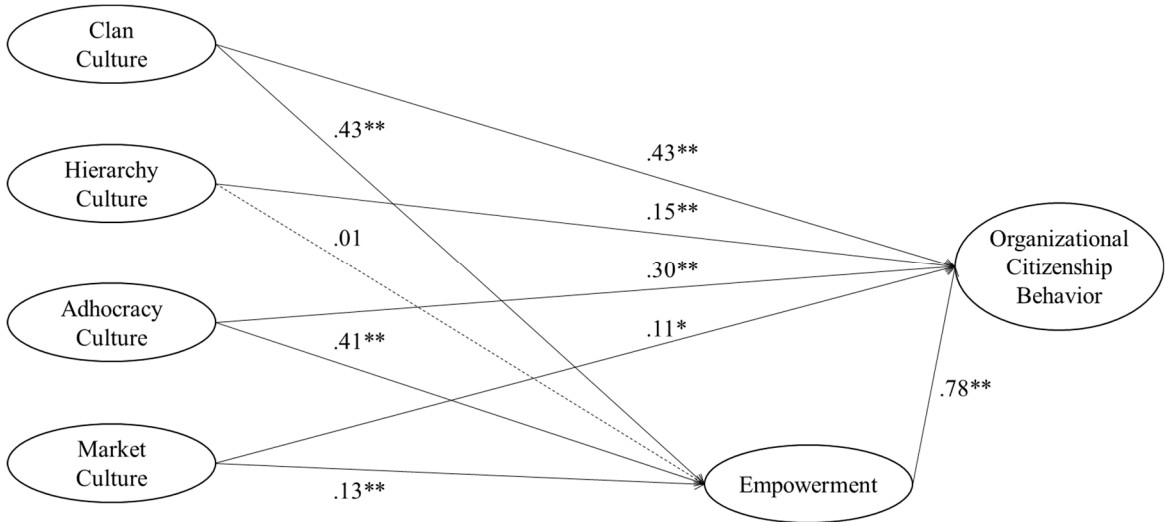

**Figure 2.** Path analysis results. Note: * *p* < 0.05, ** *p* < 0.01

To analyze the mediating effects, the current study used the bootstrap test developed by Efron and Tibshirani [74], which is a data-based resampling statistical method [75]. Of the two resampling procedures (parametric and nonparametric bootstrap testing), the current study conducted the nonparametric bootstrap testing. As shown in Table 2, empowerment partially mediates the relationship between clan culture and OCB as both direct and indirect effects are significant; therefore, H4-1 is supported. No mediation effect was found with regard to empowerment's impact on the relationship between hierarchy culture and OCB because the indirect effect is not statistically significant (*p* > 0.05) while the confidence interval includes zero points. Empowerment partially mediates the relationship between adhocracy culture and OCB as both direct and indirect effects are significant (*p* < 0.01), thereby supporting H4-3. Finally, empowerment was found to fully mediate the relationship between market culture and OCB, as the indirect effect is significant (*p* < 0.01) while the direct effect is not; thus, H4-4 is supported.

**Table 2.** The Mediating Effects of Empowerment.

| Path | | Standardized Estimate | S.E. | 95% CI (Bias-Corrected) | *p* |
|---|---|---|---|---|---|
| Clan Culture → Empowerment → OCB | Direct effect | 0.249 | 0.040 | 0.187–0.318 | 0.004 |
| | Indirect effect | 0.137 | 0.019 | 0.108–0.169 | 0.004 |
| | Total effect | 0.386 | 0.038 | 0.320–0.447 | 0.005 |
| Hierarchy Culture → Empowerment → OCB | Direct effect | 0.160 | 0.029 | 0.107–0.205 | 0.005 |
| | Indirect effect | 0.004 | 0.010 | −0.013–0.021 | 0.795 |
| | Total effect | 0.164 | 0.032 | 0.106–0.210 | 0.006 |
| Adhocracy Culture → Empowerment → OCB | Direct effect | 0.183 | 0.043 | 0.112–0.253 | 0.005 |
| | Indirect effect | 0.126 | 0.018 | 0.101–0.162 | 0.002 |
| | Total effect | 0.310 | 0.042 | 0.241–0.377 | 0.004 |
| Market Culture → Empowerment → OCB | Direct effect | 0.063 | 0.034 | 0.008–0.123 | 0.063 |
| | Indirect effect | 0.048 | 0.012 | 0.030–0.068 | 0.004 |
| | Total effect | 0.111 | 0.035 | 0.054–0.171 | 0.004 |

## 5. Discussion and Conclusions

This empirical study proposed and tested an integrated model that showed the significant influence of organizational culture and empowerment on OCB, and it also showed the mediating effect of empowerment in the Korean professional sports setting. The findings demonstrate the importance and

necessity of developing a positive organizational culture and nurturing a strong sense of empowerment in sports organizations to attain optimal organizational citizenship behavior. Where previous scholars address the importance of organizational culture on empowerment and OCB, this study reveals that different types of organizational culture have different impacts on empowerment and OCB.

From a theoretical point of view, the present study provides several contributions to research in sports organizational behavior and human resource management. First, our study investigated the impact of four different dimensions of organizational culture on empowerment and OCB, an underdeveloped area of research in sport management. Existing studies concentrate on aggregating organizational culture into a combination [76,77]. However, because organizations are composed of diverse subunits that may each have their own different cultural traits [13], it is important to study the role of each subdimension of organizational culture in understanding employees' behaviors to both academics and managers. Based on a comprehensive review of literature, this study proposed the influence of organizational culture on employees' behaviors as an aggregate as well as distinct cultural traits: clan, hierarchy, adhocracy, and market culture.

Among the four types of organizational culture, clan culture was found to most positively empower employees. Since Korea's collectivistic culture values workplace cohesion and teamwork, employees working in an organization that emphasizes a friendly atmosphere, mutual collaboration, and teamwork would allow them to be more active and competent in their assignments, thereby enabling a better outcome. This result aligns with previous literature suggesting that effective teamwork is vital for individual job performance when the specialization and complexity of task is accelerated [78], and organizations would facilitate their organizational performance through combining the thoughts, actions, and feelings of each team member [79]. This finding also supports the notion that clan culture encourages the psychological or emotional attachment of employees to the organization [80].

Furthermore, adhocracy culture was also shown to have a positive impact on empowerment. As adhocracy culture focuses on an employee's creativity, flexibility, and adaptability [13], employees who work under such a cultural atmosphere are encouraged to develop innovative and creative solutions with a high degree of empowerment. These results are also aligned with the study of Tseng [81] and Naranjo-Valencia, Jiménez-Jiménez, and Sanz-Valle [82], who highlighted the importance of a dynamic, entrepreneurial, and creative workplace that enhances employees' willingness to sacrifice themselves and take risks for the better performance and sustainability of the organization.

Interestingly, hierarchy culture was not found to be impactful of empowerment. When hierarchy culture was prevalent within an organization, the roles and responsibilities of employees were more likely to be restricted to their assigned tasks. In other words, since hierarchy culture is traditionally and implicitly embedded in most East Asian countries including South Korea [1], it is highly likely that this organizational cultural norm prevented Korean professional sport employees from actively engaging in tasks beyond their roles. Even if an organization promotes employees' confidence and creativity, employees may not feel empowered because of the deep-rooted hierarchical environment and because obedience to their supervisors is considered as an organizational virtue [1]. Finally, market culture was revealed as an important factor for enhancing employees' feelings of empowerment. As market culture emphasizes maximizing productivity and has a results-driven approach, an organization with predominant market culture usually provides as much assistance as possible to its employees to carry out maximum performance by giving them more authority [2]. In turn, empowerment plays a critical role in an organization, especially when it embraces and fosters market culture.

Second, by focusing on the relationship between organizational culture and OCB, we call for continued research on the path in various fields to future researchers. Surprisingly, in contrast to the voluminous scholarship on both organizational culture and OCB, there have been few studies on exploring the relationship between two important factors. Evidence of the path is captured by Kar et al. [33]' study that investigated the impact of components of organizational culture as antecedents of OCB and indicated that there exists a causal relationship between components of organizational culture and OCB. In other words, it is believed that employees who are properly conscious of their

organizational values tend to foster better OCB. To varying degrees, the current study shows that most sub-factors (clan culture, adhocracy culture, and market culture) of organizational culture lead to OCB, which highlights the need to incorporate organizational culture in modeling employees' OCB. Therefore, it should be reiterated that organizational culture is an indicator of OCB.

Third, this study heeds the call of existing researchers by seeking to understand the role of empowerment in the prediction of OCB. More specifically, Choi, Yoo, and Bae [83] investigated the effect of empowerment on the OCB of private social welfare organization members, suggesting empowerment has a positive effect on OCB. Shin [84] explored the impact of principals' servant leadership and empowerment perceived by teachers on OCB and found that empowerment plays a key role in forming OCB. In the sport management literature, Kim, Lee, and Won [85] examined the structural relationships between the exchange relationships, empowerment and OCB of the fitness center employees, finding that employee empowerment leads to OCB. In other words, when employees possess high levels of empowerment, they are more likely to identify themselves with the organization and became more dedicated to and involved with organizational issues [85]. Thus, the consensus appears to be that empowerment is an antecedent of OCB.

Fourth, this study contributes to sport management studies by uncovering the mediating effect of empowerment on the relationship between organizational culture and OCB. A thorough search of the literature reveals that the mediating effect has not been examined to date in the context of sport management. We examined the indirect effect of four dimensions of organizational culture on OCB. Since the indirect effect was not significant between hierarchy culture and OCB, the mediating effect of empowerment between the two was not found. However, our findings show that empowerment partially mediates the relationships between clan culture and OCB, and between adhocracy culture and OCB. In addition, it fully mediates the relation between market culture and OCB, indicating that employees have to be empowered to exercise OCB. These findings help bridge the gap in the literature by describing the detailed effects of clan, adhocracy and market culture on OCB through empowerment. The results highlight the need to include organizational culture and empowerment in models aimed at predicting sport employees' OCB.

From a practical point of view, the results offer important organizational behavior and human resource management implications for sports organizations. First, to develop clan culture, it is important to cultivate strong employee relationships. Building a strong relationship at work can improve employee satisfaction and increase their involvement in the organization; however, this kind of development takes time and effort. In order to foster strong intra-workplace connections, scheduling time for it, showing an appreciation, praising employees' performance, and showing respect to the knowledge, experience, and ability of leaders are required of all organization members. Second, to develop an adhocracy culture, managers should support employees' creativity and innovativeness [86]. Psychologically, most people tend to be uncomfortable with change because it is highly likely that employees may have to take risks and uncertainty while they shift to new ways of doing the existing tasks [87]. Nonetheless, most scholars suggest that enhancing creativity and innovation is of vital importance for deviating from conventional practices and leading organizations successfully [86]. In order to minimize the barrier of uncertainty, Hon et al. [86] suggested several human resource management (HRM) practices such as task interdependence and supportive leadership. Thus, sport managers should focus on these HRM practices to provide their subordinate with something that they need to overcome barriers or challenges when they try to innovate. Third, to develop employees' empowerment, it is imperative to inspire employee autonomy. When leaders micromanage, subordinates become passive, like bystanders rather than participants, which is ineffective and inefficient. No one likes to be considered a mere part in an organization. In order to inspire employee autonomy, leaders should trust their employees, compliment them on something they did or a choice they made, establish autonomous work teams, create decision-making opportunities, and get rid of authoritarianism.

This study also has some limitations. First, since the study was limited to male professional sports clubs, further attention should be paid to incorporating the research results into female professional

sports clubs. Future studies should expand to address this void. Second, we did not include possible exogenous variables that potentially influence organizational citizenship behavior, which may include such antecedents as organizational justice, emotional intelligence, ethical climate, organizational trust, servant leadership, and self-leadership; future studies are highly encouraged to do so. Third, this study employs empowerment as a mediator of the relationship between organizational culture and OCB. Nonetheless, based on existing studies, we believe that there are other mediating variables such as job satisfaction and organizational commitment that may influence the relationship between the two factors. Investigating other potential mediators would make the proposed framework more thorough and comprehensive.

**Author Contributions:** Conceptualization, Y.J. and J.J.Z.; methodology, Y.J.; software, M.K.; validation, Y.J. and M.K.; formal analysis, Y.J.; investigation, E.K.; resources, Y.J.; data curation, E.K.; writing—original draft preparation, Y.J. and E.K.; writing—review and editing, M.K. and J.J.Z.; visualization, E.K. and M.K.; supervision, J.J.Z.; project administration, Y.J.; funding acquisition, J.J.Z.

**Funding:** This research received no external funding.

**Conflicts of Interest:** The author declares no conflict of interest.

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
