# Peer review of "Exploring Relationships among Organizational Culture, Empowerment, and Organizational Citizenship Behavior in the South Korean Professional Sport Industry"

_sustainability, doi:10.3390/su11195412_

Round 1

Reviewer 1 Report

Overall, this manuscript is well-written drawing upon relevant theories and literature. It brings many interesting questions for the sustainability of organizations especially in the context of South Korean professional sports industry where many teams experience a shortage of financial support. I hope below comments are helpful.

Introduction

Please provide more details about how financial support from parent companies and municipal governments are used. For example, in the manuscript, the authors stated: “This has placed more burden on current employees to work extra hours without additional compensation”. More details about what percentage (and exactly what amount of financial resources, such as in USD) of financial support from parent companies and municipal governments are allocated for labor cost should be provided. (Ln 51-52) Please provide more details and/or supporting evidence why voluntary cooperation and extra-role behaviors are sometimes taken for granted (given than, ‘extra-role behaviors’ are basically something going extra miles on one’s discretion) Throughout the manuscript, please be clear about you are dealing with professional sports team employees, not professional players. Please be clear about why organizational culture matters in the context of professional sports teams.

Theoretical Background, Research Hypotheses, and Model

When theorizing, please incorporate the context & why the proposed relationships matter specifically in the context of Korean professional sports teams. Throughout the manuscript, it seems like the same hypotheses can be applied to most organizations regardless of its contexts (e.g., professional sports teams or even consulting firms, for example). (H1) Please provide separate hypotheses based on the four facets of organizational culture (i.e., clan culture, hierarchy culture, adhocracy culture, and market culture). Also, please provide a more detailed rationale about each relationship (e.g., clan culture & OCB, hierarchy culture & OCB, etc.). (H3) It is less convincing that authors propose the mediation hypothesis merely based on H1 and H2. That is, just because A->B and B->C, it does not necessarily mean B should be a critical mediator explaining the underlying mechanism through how A affect B. Hence, more theoretical rationale and supporting evidence should follow.

Method

Please provide more exact details about who did the survey. For examples, for all the measurements, did the 606 complete all the survey items? If so, please provide more details about what you did to prevent any issues related to common method bias. For self-rated items, OCB can be particularly problematic. Please provide mode details why OCB should be self-reported, not other-reported. Please provide descriptive statistics. For culture using CVM, did you aggregate the responses from participants? If so, please provide relevant statistics. Also, if so, it seems like the research questions should be analyzed in cross-level. Please provide more details regarding this point.

Reviewer 2 Report

It can be evaluated as a well-designed and organized study. Since the core of the discussion in this paper is culture, I would like you to consider once again indicators that measure culture. For this purpose, for example, the work of Geert Hofstede should be helpful. In addition, the culture and environment unique to sports organizations should be reviewed a little more. In the conceptual framework, there was no direct connection between culture and OCB, but in the SEM analysis, it seems that the discussion is not consistent. It's not bad to mix discussion and conclusion, but if possible, please separate discussion and conclusion here. In conclusion, I would like you to clarify the theoretical contribution and practical implications.

Round 2

Reviewer 2 Report

The authors responded accurately to my very difficult requests. The content is substantial and deserves to be published.

Author Response

Thank you very much for your kindly words.
